# A Case Series of 39 United States Veterans with Mild Traumatic Brain Injury Treated with Hyperbaric Oxygen Therapy

**Alison C. Bested** [1,*], **Arif M. Rana** [2], **Patrick Hardigan** [1], **Jerome Niyirora** [2], **Amanpreet Cheema** [1], **George Antony** [3], **Philip A. Defina** [4] and **Calixto Machado** [5]

1   Department of Integrative Medicine, Nova Southeastern University, Fort Lauderdale, FL 33314, USA
2   SUNY Polytechnic Institute, Utica, NY 13502, USA
3   Southeast Medical Imaging Services, Inc., Delray Beach, FL 33445, USA
4   International Brain Research Foundation, Flanders, NJ 07836, USA
5   Institute of Neurology and Neurosurgery, Havana 10400, Cuba
*   Correspondence: abested@nova.edu

**Abstract:** Importance: The Defense and Veterans Brain Injury Center reported 358,088 mild traumatic brain injury (mTBI) among U.S. service members worldwide between the years 2000 and 2020. Veterans with mTBI have higher rates of Post-Traumatic Stress Disorder (PTSD), depressive disorder, substance use disorder, anxiety disorder, and suicide than their healthy counterparts. Currently, there is no effective treatment for mTBI. Objective: To assess the efficacy of hyperbaric oxygen therapy (HBOT) as a treatment option for mTBI. Design, Setting, Participants: This is a case series of 39 U.S. Veterans diagnosed with mTBI and treated with HBOT. Of these participants, 36 were men and 3 women, and their ages ranged between 28 and 69. The treatment was administered by The 22 Project (a veteran-centered nonprofit organization) using monoplace hyperbaric chambers located in Delray Beach, Florida. Neuroimaging using Single Photon Emission Computer Tomography (SPECT) brain scans performed pre- and post-HBOT were made available for secondary analysis. Nilearn Python Library was utilized to visualize the corresponding neuroimaging data. A two-sided paired *t*-test in R was used to compare the pre- and post-treatment results. Intervention: A full treatment of HBOT involved 40 sessions. Each session consisted of the administration of 100% oxygen at 1.5 atmospheres for 90 min, twice a day, for 20 days, Mondays to Fridays only. Main Outcome and Measure: Perfusion in the brain's Brodmann Areas (BA) comparing pre- and post-HBOT using NeuroGam software analysis from brain SPECT scan neuroimaging and multi-symptom self-reports. Results: A comparison between the pre- and post-HBOT brain scans showed significant improvement in the brain perfusion, and the difference was statistically significant ($p < 0.001$). Separately, participants reported reduced pain, improved mood, and better sleep, an outcome that translated into an average of about 46.6% improvement in the measured symptoms. Conclusions and Relevance: This series demonstrated that HBOT could be a useful treatment for mTBI in U.S. veterans. The participants in the study showed marked improvement in both brain perfusion measured on SPECT scan imaging and measured mTBI symptoms. This is the first study to use brain SPECT scans with quantitative numerical measurements to demonstrate improvement in brain perfusion in veterans with mild TBI treated with HBOT and measured mTBI symptoms. Future research studies are currently being done to validate these important findings.

**Keywords:** mild traumatic brain injury; persistent post-concussion brain syndrome; hyperbaric oxygen therapy; Brodmann Areas

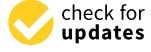



## 1. Introduction

There is an urgent need to find effective treatment options for chronic traumatic brain injury (TBI) in the United States (U.S.) veterans [1,2]. More than two million U.S. military service members have been deployed to Iraq and Afghanistan since 11 September 2001, and

approximately 10% of them have received TBI diagnoses. In about a third of the cases, TBI was due to improvised explosive device (IED) blasts [3–5]. TBI typically results from the head being struck or striking an object, the brain undergoing an acceleration/deceleration movement without direct trauma to the head (coup-contrecoup injury), or tissue pressure bubbles from blast injuries [6]. The U.S. Defense and Veterans Brain Injury Center reported that 82% of all TBIs found in 358,088 US veterans between 2000 and 2020 were mild [7]. The pathology of mild TBI consists of diffuse microscopic brain nerve cell damage that is too small to detect or diagnose using anatomical imaging such as C.T. scans [8]. However, metabolic nuclear imaging Single Photon Emission Computer Tomography (SPECT) scans that use a radioactive tracer and a camera to create 3-D images can document blood circulation inside the brain and detect microscopic injuries associated with mTBI [9,10]. A review of imaging studies [11–13] show that SPECT is more sensitive than C.T. scans in detecting lesions in TBI patients. SPECT appears to be better than CT or MRI in determining long-term prognosis. Currently, functional MRI represents a very promising technology for prognosticating cognitive recovery after TBI, but further rigorous testing is still required before such results can be used clinically [14].

The criteria for mild TBI (mTBI) include a confused or disoriented state that lasts less than 24 h, loss of consciousness for up to 30 min, and/or memory loss lasting less than 24 h [7]. Other cognitive difficulties include feeling dazed, groping for words, and having problems with attention, concentration, processing speed, planning, reasoning, and abstract thinking. Physical symptoms include headache, nausea, vomiting, drowsiness, fatigue, dizziness, loss of balance, blurred vision, unrefreshed sleep, sensitivity to light or sound, tinnitus, and alteration of taste or smell. Some patients also experience mental health symptoms including anxiety, depression, post-traumatic stress disorder (PTSD), and suicide ideation [15]. Suicide rates, which increase with more TBIs, are two times higher in mTBI veterans compared to their healthy counterparts [16–18]. In 2012, it was estimated that 22 U.S. veterans died from suicide each day [19].

Current treatment modalities of mTBI focus on pharmaceutical and psychological therapies for mental health symptoms. This treatment approach has been mildly to moderately effective at best in alleviating symptoms such as headaches, sleep disturbances, mental health disorders, cognitive difficulties, and fatigue [20–24]. The lackluster success of these therapies is likely attributable to the absence of specific treatment options to address the underlying biological mechanism of brain cell injuries in mTBI patients [25,26]. A possible new treatment option for mTBI is hyperbaric oxygen therapy (HBOT). HBOT has been shown to promote neurogenesis, angiogenesis, cellular metabolism, and mitochondrial function [27]. HBOT also reduces inflammatory cytokines, inflammatory reactions, brain edema, apoptosis, and oxidative stress [28–42]. There is increasing evidence suggesting that HBOT improves post-concussion syndrome and PTSD after moderate brain injury, and HBOT effectively reduces chronic post-traumatic anxiety and suicide [43–47]. Presently, the U.S. Food and Drug Administration approves HBOT to treat decompression sickness (BENDS), infections such as refractory osteomyelitis, diabetic wounds, carbon monoxide poisoning, air/gas embolism, acute traumatic ischemia, radiation injury, sensorineural hearing loss, and severe blood loss [48,49].

## 2. Methods

Researchers from the Institute of Neuro-Immune Medicine (INIM) at Nova Southeastern University (NSU) in Davie, Florida, collaborated with The 22 Project, a Veteran-centered family foundation, to evaluate the effectiveness of HBOT in treating mTBI in 39 U.S. veterans. The 22 Project foundation administered HBOT and gave the pre- and post-treatment brain SPECT scans to NSU researchers for secondary analysis. Brain SPECT scans were provided by The 22 Project from the veterans who had completed The 22 Project HBOT program for mTBI between 1 March 2016 and 5 March 2019. An infusion of 30 mCi of Tc99m-Ceretec was infused through the veteran's IV site while the veteran was performing a Stoop Test. Brain SPECT scans were conducted 40–60 min after the injection of the

radiotracer by placing the veteran under the gamma scintillation camera (Siemens E-cam Dual Head) with a low-energy, high-resolution collimator. Visual analysis was conducted comparing the pre- and post-treatment studies that were normalized to cerebellum brain activity. SPECT images were reoriented into Talairach space using NeuroGam software (Segami Corporation) for the identification and numerical calculation of the mean perfusion value of each of the Brodmann Areas. In addition, volume-rendered brain perfusion images normalized to cerebellum maximal activity were reconstructed. The change in BA perfusion for each subject was determined by calculating the percentage difference between post-period and pre/baseline-period divided by the pre/baseline-period perfusion. This case series was the first to have SPECT scans with Neurogam quantitation available for analysis. This formed the present case series of 39 veterans which was not a research study and did not have a control group. Each veteran had different brain injury locations. The injuries were reflected by reduced perfusion values in the individual pre- and post-BAs. Pre- and post-HBOT SPECT scan perfusion values for the BAs were compared for each individual veteran. Because we only had pre- and post-HBOT brain SPECT scans, no multiple comparisons could be done. Due to the different brain injury locations, no correlation or multiple comparisons could be made among the perfusion values of the veterans' BAs.

The NSU Institutional Review Board reviewed and exempted this study. This study complies with the Strengthening the Reporting of Observational Studies in Epidemiology (STROBE) reporting guideline.

### 2.1. Study Population

Participants (Figure 1) were recruited through The 22 Project's website and selected on a first available treatment basis. The main eligibility criterion was to be a U.S. veteran who has been diagnosed by self-report with chronic mTBI (between 3 to 10 years after their mTBI event). There was no specific information available for the individual veteran. Participants were screened for smoking/vaping for 2 months and any substance abuse for at least one year. If they screened positive, the veteran was placed on a future list and asked to reach out when they had stopped smoking or drinking etc. Before actual treatment, participants were screened by the hyperbaric oxygen facility using the established exclusion criteria for HBOT. The absolute contraindication for HBOT is an untreated collapsed lung. Relative contraindications include pulmonary diseases (chronic obstructive lung disease, which includes bullae/cyst, upper respiratory infections), recent ear or thoracic surgery, uncontrolled fever, claustrophobia, use of the medications bleomycin or doxorubicin, autoimmune diseases, cancer (except localized skin cancer), diabetes (well-controlled is acceptable), heart disease (congestive heart failure, controlled hypertension is acceptable), infectious disease lasting six months, kidney disease, liver disease, neurological disease (Alzheimer's disease, multiple sclerosis, stroke), bipolar disorder, untreated depression or schizophrenia, alcohol abuse, drug abuse, and smoking [29]. There was no known bias in the selection process, and participation in the study was voluntary. All of the screened veterans using the exclusion criteria were able to undergo HBOT.

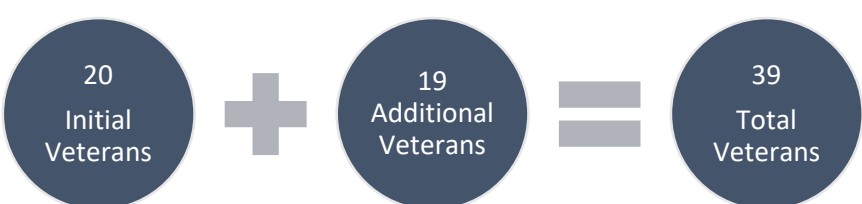

**Figure 1.** Study population flowchart.

### 2.2. Overall Approach

Henry's Gas Law provides the foundation for investigating the potential benefits of using HBOT to treat mTBI [50]. This law states that the solubility of gas within liq-

uid increases as the external pressure inside a closed chamber is increased at a constant temperature (Figure 2).

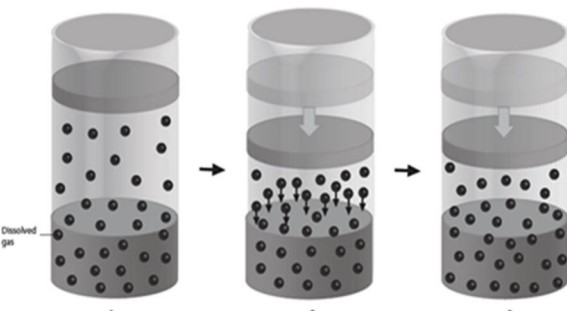

**Figure 2.** Henry's Gas Law—(1) Normal pressure with gas molecules in equilibrium with the liquid inside a closed container. (2) Increased pressure forces gas molecules into the liquid inside a closed container. (3) New equilibrium with increased gas/oxygen molecules within the liquid/plasma inside a closed container/HBOT chamber with increased pressure maintained.

Applying Henry's Gas Law to HBOT implies that the amount of gas (oxygen) dissolved in a liquid (plasma) increases when the pressure inside a closed HBOT chamber is increased. In this study, the pressure was increased from 1 atmosphere (ATM) to 1.5 ATMs. That is a 50% increase in the oxygen dissolved in the plasma, which is an effective treatment of its own merit. It was anticipated that HBOT would cause cellular repair by increasing oxygen carried in the plasma to the brain tissues. The difference in tissue–plasma gradient was expected to cause the oxygen to diffuse from the plasma into the damaged brain tissues [51–53]. By applying Henry's Gas Law, it logically follows that using room air inside a pressurized HBOT chamber cannot serve as sham control for HBOT. Oxygen is increased in the plasma, even when only room air is used. Therefore, room air under pressure also raises the oxygen available to the tissues and is a therapeutic treatment and not a sham [54,55]. To avoid the flaw in the sham control design with HBOT, a crossover research study design can be performed in the future to test the neuro-therapeutic effects of HBOT [27,56–61]. This was a case series and there was no control group for comparison.

A complete regimen of HBOT treatment for each participant was 40 sessions. Each session consisted of the administration of 100% oxygen at 1.5 ATMs for 90 min, twice a day, for 20 days, Mondays to Fridays only. Any two given sessions were separated by a 3 to 4 h break. On day 1, before the start of HBOT sessions and after the 40th treatment, each veteran had brain SPECT scans taken to acquire colorimetric images depicting perfusion in the brain's Brodmann Areas (BA). BA regions correspond to specific brain functions such as memory, attention, emotion, etc. [62]. An infusion of 30 mCi of Tc99m-Ceretec was infused through the veteran's IV site while the veteran was performing a Stoop Test. As noted previously, brain SPECT scans were conducted 40–60 min after the injection of the radiotracer by placing the veteran under the gamma scintillation camera (Siemens E-cam Dual Head) with a low-energy, high-resolution collimator. Visual analysis was conducted comparing the pre- and post-treatment studies that were normalized to cerebellum brain activity. SPECT images were reoriented into Talairach space using NeuroGam software (Segami Corporation) for the identification and numerical calculation of the mean perfusion value of each of the Brodmann Areas. In addition, volume-rendered brain perfusion images normalized to cerebellum maximal activity were reconstructed. The change in BA perfusion for each subject was determined by calculating the percentage difference between post-period and pre/baseline-period divided by the pre/baseline-period perfusion. An average of these perfusion changes for each BA was calculated [63]. The perfusion scale is rated on a scale of −5 to +5, −5 being least perfused and +5 being most perfused. On a SPECT image, the least perfused BA are dark blue or black in color (Figure 3).

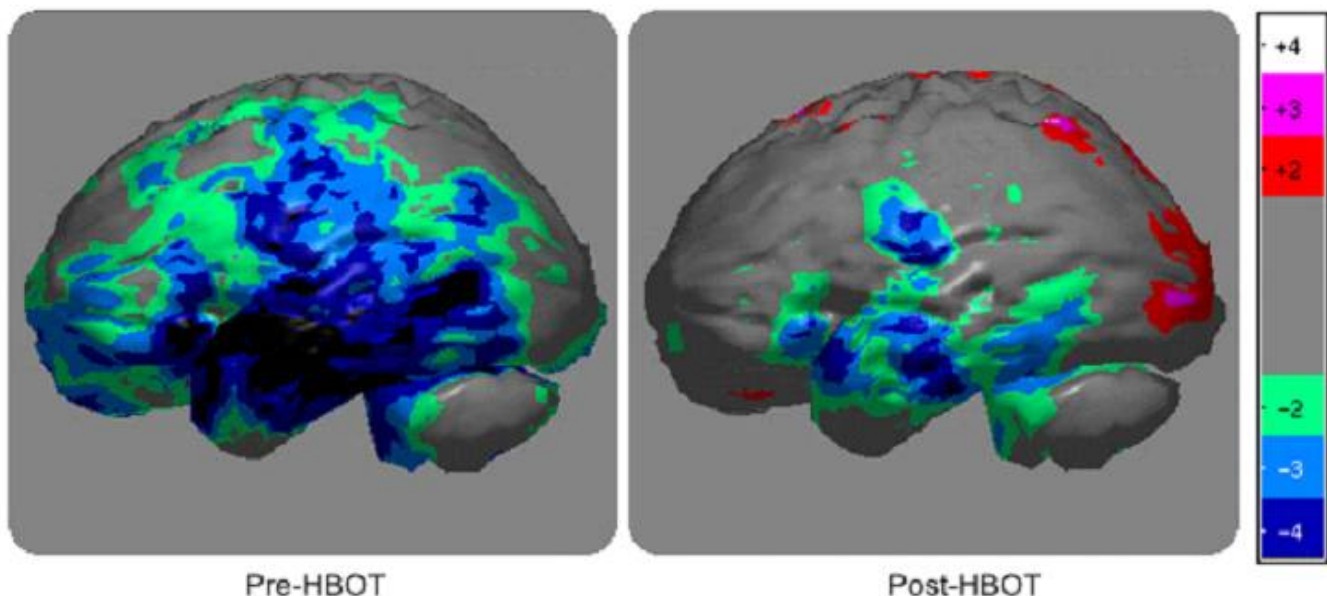

**A**

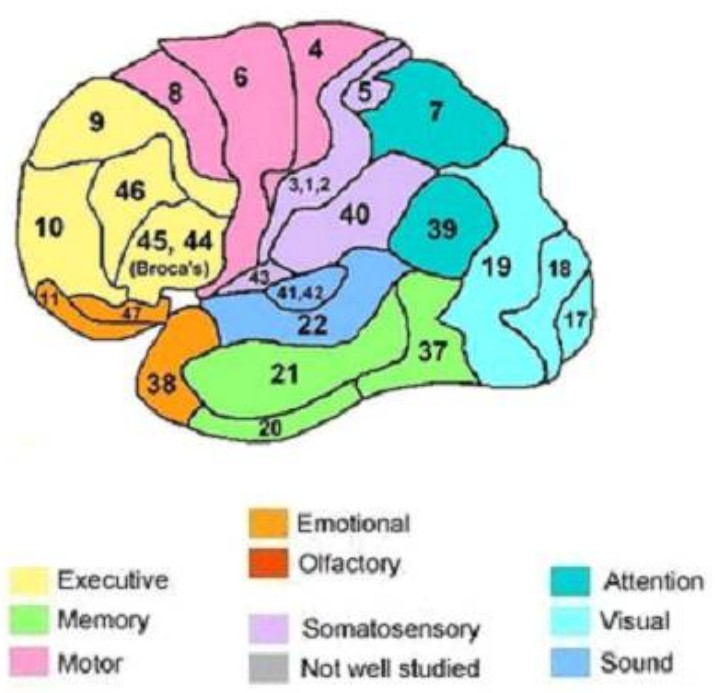

**B**

**Figure 3.** (**A**) Brain SPECT Scan Image with Colorimetric Scale: Grey—normal blood flow/perfusion. Light Green—slightly reduced perfusion. Light blue—moderately reduced perfusion. Dark blue/black—severely reduced perfusion. The Veteran's brain SPECT scan illustrates the changes in color in the temporal lobe of the brain after HBOT from dark blue to light blue and green colors. (**B**) Brodmann Areas with functional areas of the brain.

The 22 Project conducted all treatments in monoplace HBOT chambers located in Delray Beach, Florida. Each participant was monitored until the end of HBOT treatment, and no side effects were reported. Post-treatment, participants completed a multi-symptom

questionnaire (MSQ), a self-report questionnaire that measures sleep, anxiety, depression, pain, and fatigue [64]. The 22 Project provided NSU researchers with de-identified SPECT scan data corresponding to 39 participants chosen randomly (36 men and 3 women, aged between 28 to 69 years) for a secondary analysis about the effectiveness of HBOT in treating mTBI.

### 2.3. Statistical and Data Analysis

The numerical data from the SPECT scans were extracted using the NeuroGam software and preprocessed in an Excel spreadsheet. None of the participants had missing data. The pre- and post-treatment mean perfusion scores were calculated for each BA. These scores were then plotted on the brain model (Figure 4) using the Nilearn Python Library [8,65].

Before treatment

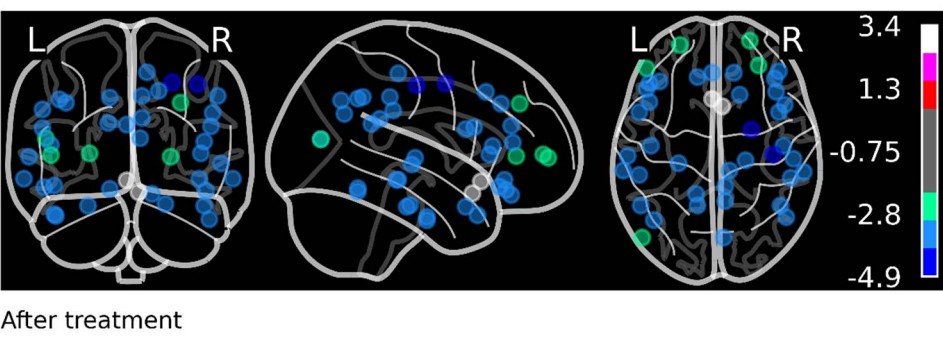

After treatment

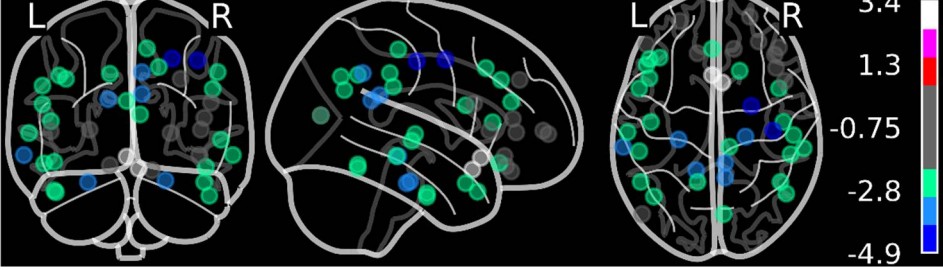

**Figure 4.** The pre- and post-HBOT mean perfusion levels from all veterans were calculated for each Brodmann area and plotted using the Nilearn Python Library. The Colorimetric Scale shows improvement in perfusion from blue to green color after HBOT.

The 3D brain coordinates of each BA were obtained from [66]. For better visualization, a random jitter number was added to each coordinate point to minimize the overlapping of the points. On a colorimetric scale, the minimum perfusion score was −4.9 (dark blue color), whereas the maximum score was +3.4 (white). Figure 5 shows the numbers of BA that improved (green color) and those that did not improve (red color) for each individual veteran.

The distribution and dispersion of the perfusion scores were examined through descriptive numerical summaries and graphical tools in R. Using two-sided paired *t*-tests, we compared the change in blood flow levels pre- to post-HBOT. All hypothesis testing was carried out at the 5% (two-sided) significance level. *p*-values were rounded to three decimal places. *p*-values less than 0.001 were reported as <0.001 in tables. *p*-values greater than 0.999 were reported as >0.999. R (version 4.2.0) statistical package was used for descriptive calculations and *t*-tests. Due to insufficient data, no subgroup examination or sensitivity analysis was conducted.

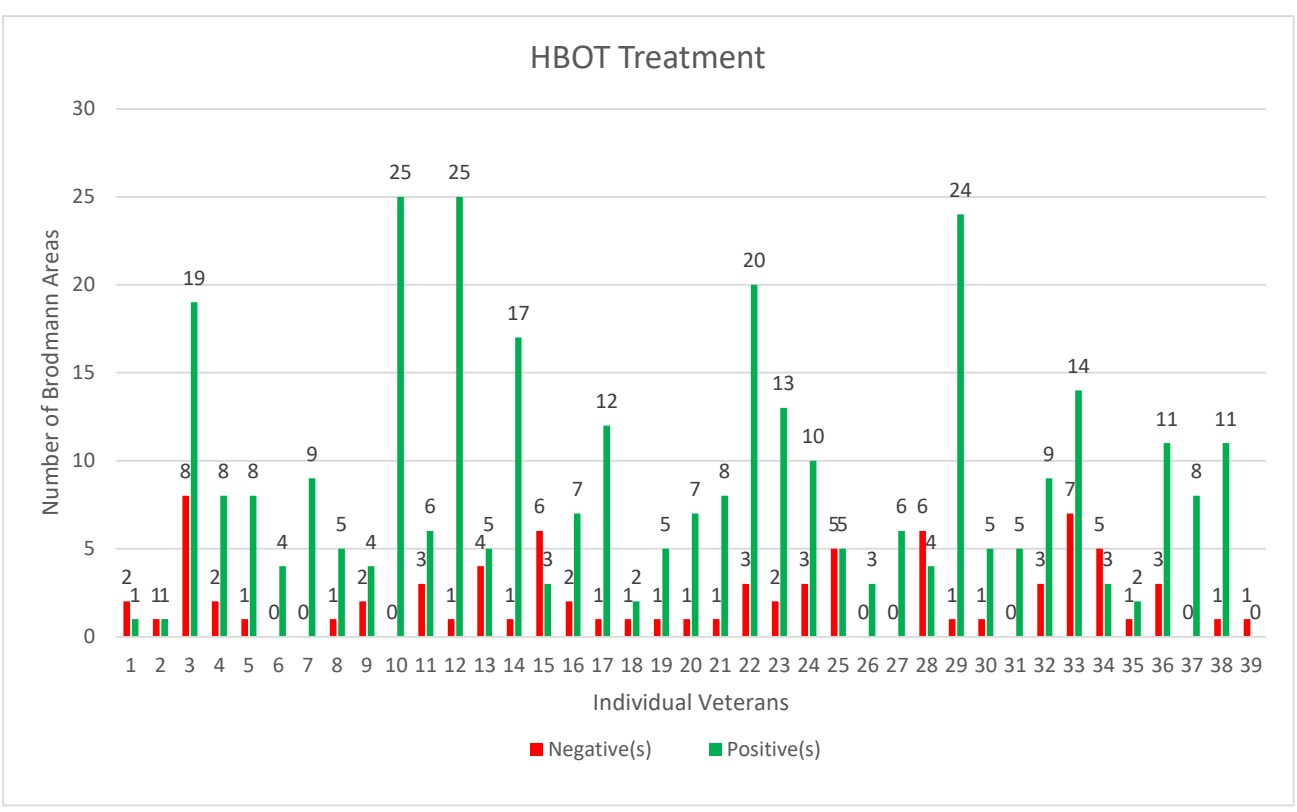

**Figure 5.** Individual Veteran's BA after HBOT (green improved BA, red not improved or worse BA).

### 3. Results

Initially, 20 SPECT scans of participants with mTBI, provided by The 22 Project, were analyzed. Based on the colorimetric changes of the SPECT scans, with the color improving from dark blue to light green, it was observed that there was a significant improvement in brain perfusion levels post-HBOT. At the same time, the MSQ average score from these veterans was consistent with a 46.6 percent improvement in all symptoms measured post-HBOT (Table 1 and Figure 6). A paired *t*-test was performed (Table 2) using SPSS (version 28.0.1.0) and results indicated that there was a difference between the initial and the 40th treatment ($p < 0.001$).

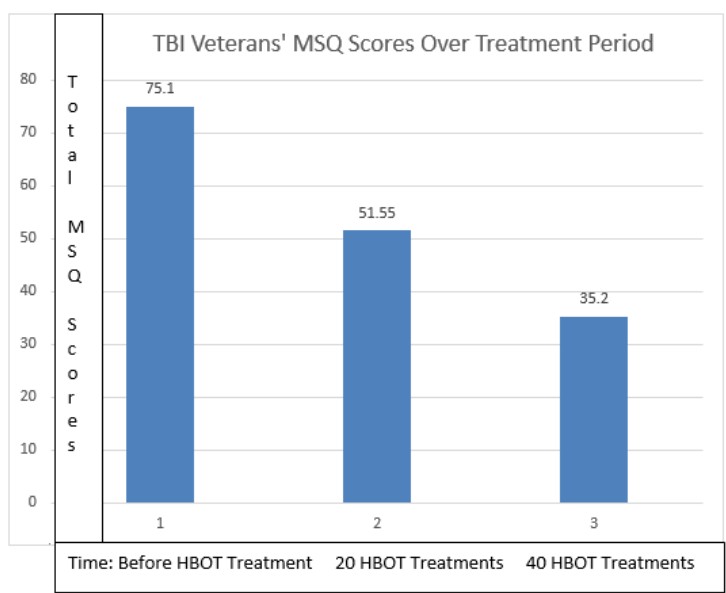

**Figure 6.** Symptom improvement was 46% comparing pre- and post-HBOT.

**Table 1.** Total MSQ Scores for 20 veterans pre- and post-HBOT.

| Veteran | Initial MSQ | 20th Tx MSQ | 40th Tx MSQ | Last Tx |
|---|---|---|---|---|
| 1 | 107 | 77 | 56 | 8/8/2017 |
| 2 | 26 | 25 | 14 | 10/12/2018 |
| 3 | 85 | 21 | 11 | 5/1/2018 |
| 4 | 126 | 57 | 41 | 11/17/2017 |
| 5 | 129 | 67 | 21 | 2/8/2017 |
| 6 | 110 | 73 | 40 | 9/21/2017 |
| 7 | 41 | 24 | 17 | 2/16/2017 |
| 8 | 65 | 34 | 10 | 11/30/2017 |
| 9 | 94 | 38 | 16 | 8/20/2018 |
| 10 | 64 | 63 | 42 | 12/27/2018 |
| 11 | 15 | 24 | 16 | 2/4/2019 |
| 12 | 33 | 21 | 19 | 2/25/2019 |
| 13 | 21 | 64 | 23 | 10/12/2018 |
| 14 | 101 | 73 | 55 | 10/15/2018 |
| 15 | 37 | 39 | 37 | 3/22/2019 |
| 16 | 160 | 154 | 154 | 3/20/2019 |
| 17 | 106 | 49 | 31 | 3/25/2019 |
| 18 | 56 | 36 | 27 | 4/23/2019 |
| 19 | 97 | 70 | 57 | 4/24/2019 |
| 20 | 29 | 22 | 17 | 5/22/2019 |
| **Mean** | 75.1 | 51.55 | 35.2 | |
| **Std. Deviation** | 42.0224251 | 31.47175 | 31.86088 | |

**Table 2.** Paired *t*-test between initial MSQ and 40th Tx MSQ.

| | | **Paired Samples Test** | | | | | | | | |
|---|---|---|---|---|---|---|---|---|---|---|
| | | **Paired Differences** | | | | | | | **Significance** | |
| | | **Mean** | **Std. De-viation** | **Std. Error Mean** | **95% Confidence Interval of the Difference** | | **t** | **df** | **One-Sided *p*** | **Two-Sided *p*** |
| | | | | | **Lower** | **Upper** | | | | |
| Pair 1 | Initial MSQ—40th Tx MSQ | 39.90000 | 33.16768 | 7.41652 | 24.37705 | 55.42295 | 5.380 | 19 | 0.000 | 0.000 |

Neuropsychological assessments were only available on 4 of the 20 veterans. The Beck Depression Inventory I-II showed improvements in the veterans' depression scores. The State-Trait Anxiety Inventory showed a decrease in anxiety scores. SPECT scans from 19 additional veterans were added, and all were analyzed. The mean perfusion value for each of the Brodmann Areas was assessed. The quantitative scale for the brain SPECT scans ranged from −5 to +5 on the colorimetric scale and also numerically on the table generated by the Neurogram software for each of the BA. The pre- and post-HBOT values were compared for each BA. Any value below −3.00 was compared pre- versus post-HBOT. The value of −3.00 was selected as it indicated a severe perfusion deficit on the SPECT scans compared to normative data [67] and was seen as dark blue on the colorimetric scale of the brain SPECT scan. An improvement in the BA was defined as a 10% positive change in the mean perfusion value and based on the colorimetric scale seen the brain SPECT scan. We correlated the change in the Colorimetric Scale with the quantitative data from the NeuroGam software. A change of 10% was easily distinguishable. A marked improvement in the brain perfusions was observed. As portrayed in Figure 4, before treatment, the severely affected BAs were dark blue in color. After treatment, the color changed to light green or grey in most BA, showing the improvement in brain perfusion after HBOT.

Figure 5 validates that there were more improved BA (green color) than not-improved (red color) for most veterans. Using a two-sided paired *t*-test, the difference in perfusion in

each veteran's Brodmann Areas that had reduced blood flow (pre- vs. post-HBOT) was found to be statistically significant at a *p*-value < 0.001 (Table 3). This table compared the mean value of all of the BA from the Neurogam quantitation before and after HBOT. The affected Brodmann Areas were different for each veteran due to the specific location of their brain injury.

**Table 3.** Summary of Statistics Pre- and Post-HBOT.

|  | Pre-HBOT | Post-HBOT | Change | *p* Value |
|---|---|---|---|---|
| Brodmann Area Mean Value | −1.65 | −1.35 | 0.29 | <0.001 |
| Std Dev | 1.45 | 1.46 | — | |
| Std Err Mean | 0.03 | 0.03 | 0.01 | |
| Upper 95% Mean | −1.60 | −1.30 | 0.32 | |
| Lower 95% Mean | −1.70 | −1.41 | 0.26 | |
| N | 2904 | 2904 | | |

While there was no formal follow-up of participants post-HBOT, The 22 Project kept in touch informally with approximately 50% of their veterans who reported symptoms improvement. The veterans reported their symptoms had not regressed and none of the treated veterans had committed suicide.

## 4. Discussion

A significant number of U.S. military service members who were deployed to Iraq and Afghanistan had post-mTBI symptoms, a condition that is associated with a multitude of cognitive, physical, and mental health symptoms. Current treatment modalities for mTBI that focus on pharmaceutical and psychological therapies for mental health symptoms have largely failed to yield sustained improvement in veterans' symptoms. In this study, we investigated whether HBOT could be an additional treatment for mTBI. To this end, we collaborated with The 22 Project, a Veteran-centered nonprofit organization, to assess the efficacy of HBOT as a treatment option for mTBI. The 22 Project recruited and administered HBOT treatment to 39 U.S. veterans who suffered from mTBI. Of note, each veteran had different locations of brain injuries that were reflected by reduced perfusion values in different BAs. Pre- and post-HBOT SPECT scan perfusion values were compared for each individual veteran. Due to the different brain injury locations, no correlation could be made among the perfusion values of the veterans' BAs. Each participant received 40 sessions of HBOT. One brain SPECT scan was done pre- and post-HBOT. The 40 HBOT sessions and 1.5 ATM were based on a study by Paul Harch [68]. Data analysis suggested improvements in most participants. The difference in perfusion (pre- vs. post-HBOT) was statistically significant (*p*-value < 0.001. This improvement in brain perfusion coincided with a self-reported improvement in mTBI symptoms from the initial group of 20 veterans. These veterans reported better sleep, reduced pain, and improvement in mood. None of the veterans committed suicide during this study. These findings are consistent with previous studies that suggested the efficacy of HBOT in reducing chronic post-traumatic anxiety and suicide and alleviating post-concussion syndrome and PTSD symptoms after moderate brain injury [26–30]. The brain has the ability to heal and repair itself if the correct environment is present. The presence of increased oxygen availability in the brain tissue is a result of HBOT therapy. HBOT has two components: (1) increased oxygen and (2) increased pressure in a closed system—the blood vessels. Oxygen flows from the plasma, located inside blood vessels, into the injured brain tissue due to the difference in the oxygen gradient. Oxygen present in high levels within the plasma diffuses into the injured hypoxic brain tissue which has lower oxygen levels. The presence of oxygen in the injured brain tissues causes the reduction of cerebral ischemia, improves cerebral metabolism by increasing mitochondrial function, decreases cerebral vessel spasm, improves the collateral

circulation via angiogenesis, signals growth factors for repair, and stimulates brain stem cells to replace poorly functioning or missing nerve cells. It also reduces inflammation by decreasing pro-inflammatory cytokines and increasing anti-inflammatory cytokines. These cellular repair and wound-healing mechanisms are a result of increased oxygen availability in the local hypoxic brain tissues [69,70].

Previous studies by Ma, Harch, and Hart [71–73] showed improvements using changes in symptoms, neuropsychological testing, and perfusion MRI to compare pre- and post-HBOT in mTBI. This is the first study to use brain SPECT scan with quantitative numerical measurements to compare the effects on symptoms and brain perfusion in veterans with mild TBI treated with HBOT.

Accordingly, we conclude that HBOT is a viable treatment option for mTBI that warrants additional studies to verify the findings from this study. We have obtained funding for a pilot veteran study which will repeat this study and add symptom questionnaires and markers of inflammation (cytokines and mitochondrial function) pre- and post-HBOT.

## 5. Limitations

The sample size in this case series was relatively small, and no information was available about the date of injury with mTBI diagnosis. Veterans self-reported their mTBIs. Retrospective self-report, especially if more than a decade post-injury, may be unreliable. Future studies should use doctor-diagnosed assessments or documentation of mTBI within theatre. There was no control group in this case series and no control for multiple comparisons. Neuropsychological testing ideally should be done on all the veterans, pre- and post-HOBT. In future HBOT studies, data analysis to assess the correlation between the MSQ symptoms and the post-SPECT scan data would be beneficial. Only the total scores and not the individual MSQ question scores of the veterans were available for this case series. Therefore, we could not correlate the BAs and the MSQ scores. Additionally, participants were obtained from convenience sampling. Since it is not possible to have a sham study with HBOT due to the application of Henry's Gas Law, a larger crossover study is warranted for future research studies. Veterans exposed to numerous neurotoxicant exposures, most prominently burn pits, can develop serious lung conditions following these exposures. This group of veterans would not be able to undergo HBOT. Additionally, a mechanism to follow the veterans' symptoms post-HBOT is needed to better understand the long-term effects of the treatment.

## 6. Conclusions

This case series demonstrated that HBOT is an effective treatment for mTBI/persistent post-concussion syndrome in U.S. veterans. The participants in the study showed marked improvement in both brain perfusion studies and measured mTBI symptoms. The veterans reported better sleep, reduced pain, and improvement in mood post-HBOT. Other studies have reported the effectiveness of HBOT in treating symptoms similar to those of mTBI, such as PTSD after a brain injury. More studies are needed to confirm that HBOT is an effective long-term solution to treat mTBI, improve the quality of life for thousands of U.S. veterans suffering from this debilitating condition, and stop the 26 US veterans with mTBI from committing suicide every day.

**Author Contributions:** Conceptualization, A.C.B.; formal analysis, P.H. and J.N.; investigation, A.M.R. and P.A.D.; methodology, A.C.B., A.C. and P.A.D.; project administration, C.M.; resources, A.M.R. and G.A.; supervision, C.M.; writing—original draft, A.C.B.; writing—review & editing, A.C.B., A.M.R. and C.M. All authors have read and agreed to the published version of the manuscript.

**Funding:** Funding/financial support was provided by The 22 Project, a Veteran-centered family foundation located in Delray Beach, Florida. The EIN (non-profit) number for The 22 Project is 47-1180415.

**Institutional Review Board Statement:** The study was conducted in accordance with the Declaration of Helsinki, and approved by the Institutional Review Board of Nova Southeastern University (IRB #:

2021-512; Title, "A Case Series of 39 United States Veterans with Mild Traumatic Brain Injury Treated with Hyperbaric Oxygen Therapy; 28 October 2021).

**Informed Consent Statement:** Informed consent was obtained from all subjects involved in the case series.

**Data Availability Statement:** The data presented in this study are available on request from the corresponding author. The data are not publicly available due to privacy restrictions.

**Acknowledgments:** The authors wish to thank Alex Cruz, Erica Cruz, Connie Governale, John DeLuca, Nancy Klimas, Sonia Neubauer, Beth Gilbert, and Louisa Visconti, for their support and feedback on this case series.

**Conflicts of Interest:** The authors declare no conflict of interest.

**Role of the Funder/Sponsor:** No involvement in the study design, data collection, data analysis interpretation of the results, preparation of the manuscript, or in the decision to publish the findings.

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
