# Peer review of "A Case Series of 39 United States Veterans with Mild Traumatic Brain Injury Treated with Hyperbaric Oxygen Therapy"

_ctn, doi:10.3390/ctn6030021_

Round 1

Reviewer 1 Report (New Reviewer)

This study presents brain spect and patient reports from a series of 39 veterans exposed to 40 twice per day HBO2 sessions.

I offer a number of comments:

1. The word "efficacy" is used in this paper. This word is incorrectly used. Efficacy refers to phase III randomized trials.

2. The word "effectiveness" is used in this paper. This word is incorrectly used too. This word refers to phase IV trials. 

I ask if the authors are certified and up to date with CITI and GCP.

3. I ask for the formal study protocol to be submitted to this journal.

4. A few symptoms are reported. Much is written in this paper about spect imaging.  I suggest a complete revision. The clinical expression and complaints are more compelling than some sort of brain imaging. Alternatively, you could use the spect results to make hypotheses about what HBO2 might be doing, but to draw the conclusion that Veterans with mTBI should receive 40 treatments with HBO2 is not supported well by this data. 

5. What is the MSQ?  Please add it to the paper. I looked for this measure online and could not find it. Has it been standardized? Please provide details.

6. The paper does not provide tabular details about the Beck and State Trait tests. Were these done in all 39, or just in 4 of the 20?  And why 20?  Your study was about 39.  Adding a flow diagram showing numbers of participants and what tests they had or not, would be helpful.

7. The paper uses the term "suffer from mTBI." I understand what you mean, but it would be preferable and more precise to say something like post-mTBI symptoms, or post-concussive syndrome (if they meet the DSM-V definition), etc.

8. There is no description about these participants. What symptoms did they have?  What was their neurological exams?  What were their injuries? Did you administer the OSU to document their history of brain injury in each?  Is so, please report. If not why not? The paper presents very little confidence that these participants even met the DVBIC definition of mTBI, and if so, how many TBIs had they had?

9. In the introduction you mention the urgency of helping veterans. I think non-veterans deserve help too.

10. Why did you chose BID HBO2 rather than once per day. Having only 1.5 hours between exposures seems way too short. Why did you chose this?

11. Since much is presented about SPECT...I find it difficult to accept that the IRB exempted this study. What was the SPECT radiation dose?  Generally it is at least 12 mSv per scan, or more. Was the IRB aware that each participant would receive that much radiation?  I think you should submit your formal review and exemption letter to the editor in this regard.

Next, the IRB should only exempt a study that is no more than minimal risk: essentially no more risk than daily life. Exposure to hyperbaric oxygen should not be exempt by an IRB. Again, I recommend that the IRB correspondence be submitted to the Journal.

12. What does a normal SPECT look like?  Are there studies of SPECT scans in normal, healthy people receiving the Stroop? 

13. Are you sure this is the first study showing quantitative changes in SPECT?  I thought Dr. Harch and Dr. Efrati had done something similarly.

14. The paper says that mild TBI have normal CT scans. Generally true, but there is a sub-category called mild-complicated in which the clinical diagnosis is mild TBI, but the CT is abnormal, hence this term. At the time of mTBI were CT scans done of your group? If so, do you have the results? If you do, did anyone meet the complicated diagnosis?  Also, in this population many, if not all likely had more than one TBI, including potentially more than one mTBI. The paper does not inform the readers about the TBI history, but should. Did no one not take a detailed history?

15. The paper says that the FDA approves hyperbaric oxygen.  This is inaccurate. The FDA accepts the older UHMS list of approved disorders, but there is not a single HBO2 indication that is FDA approved with a label. The reason for this, the UHMS list existed before the existence of the FDA. For example, acute hearing loss is not on the FDA approved list and likely never will be as the FDA now requires at least one phase III efficacy trial before granting a label.

16. The paper says group comparisons cannot be offered because of heterogeneity. Can you not somehow merge the pre- and post -spect results to derive this?  Perhaps a medical statistician would be helpful in this regard.

17. Study inclusion is just someone saying they had a TBI and now have problems (which are not articulated in the paper). The lack of explicitness is very problematic and needs to be corrected. If you cannot, that is a major limitation.

18. Your HBO2 exclusion list omits heart failure, lung cyst or bullae.

19. The section about Henry's law can be omitted.

20. The paper says sham chamber sessions cannot be done. This is incorrect. They can be done. You might want to go to clinicaltrials.gov and look at the protocol by Dr. Efrati's group about HBO2 for post-covid problems, or Weaver's study of HBO2 for brain injury on clinicaltrials.gov where both studies used a very low pressure of air to provide an effective sham. There are other shams that have been done, but you argue they offer biological effect so are inappropriate. Ok I suppose. In that case why did you not at least do a randomized HBO2 v no HBO2 trial much like Efrati's group did with mTBI?

The strength of inferences that one can draw from not having a sham, or even a randomized non-HBO2 controls is weak compared to blinded and sham-controlled.

21. What was the O2 flow through the chamber?  What chamber size?  This is important as it takes time to displace nitrogen from the chamber. If chamber O2 flows were low, and if the chamber is large, it takes quite a long time before the inspired O2 level exceeds 95%.

22. What was the chamber compression and decompression times?  Was this standardized? 

23. What were adverse events, including not related to spect or to HBO2? Trials should list these.

24. It does not seem that reference 61 has anything to do with symptom reporting.  Please confirm.

25. In the paragraph just before the Results the paper talks about comparing changes in oxygenation. This is not what this study did. It compared pre to post information regarding a 40- course of bid HBO2.

26. In the Results, I do not understand the additional 19 to the 20. Please make all this clear.

27. Why did you not do formal 6 and 12 month or longer follow up?  Are you not interested in learning if what you did had durable results? You say approximately 50% -- well, was it 49%, 51%?  What was it, exactly? How long after HBO2 was this informal contact?

I think your sentence, "none of the treated veterans had committed suicide" should be changed; perhaps: "none of the veterans in which we had follow-up (n=??) exposed to 40 HBO2 sessions attempted suicide." 

28. The Discussion talks about many aspects of what HBO2 might do, but largely not referenced. For example, where is the evidence that HBO2 increases mitochondrial function in people suffering sequelae after TBI? Typically HBO2 causes vasoconstriction, at least at higher doses of O2 than used in your study. Anyway, many potential effects are listed, but none are known to occur in patients like you exposed to HBO2. 

29. I would hope the study you refer to happening after funding is obtained overcomes many of the problems with this study.  Having an experienced clinical trial investigator is advisable.

Author Response

Please see comments in the attached cover letter.

Reviewer 2 Report (New Reviewer)

The manuscript is interesting and well written. It is acceptable for publication 

Author Response

Thank you for your support and feedback.  We have made some additional corrections in the revised paper.

Reviewer 3 Report (New Reviewer)

Dear colleagues

Your study is very interesting.

However, I would make a few remarks:

- Figure 1 and Table 2 contribute nothing to the overall understanding. indeed, it seems to me useless to illustrate Henry's law, just like the T-test which is well described in the text.

- Figure 4 is very difficult to read. it should be reworked.

- why do you speak of 20 SPECTs when figure 4 shows 39 results? there are only 4 patients evaluated in neuropsychology? you should clarify the situation with a flow chart in chapter 2.1 study population

Best wishes

Author Response

Dear Reviewer, thank you for your feedback.  We have addressed the following concerns you had:

1) Figure 1 has been changed to Figure 2 and this figure was kept because understanding Henry’s Gas Law is essential when using HBOT as a therapy to induce healing of injured brain tissues. 

2) Figure 4 has been changed to Figure 5 and we have enlarged it for better readability.

3) Figure 1 has been added.  The process began when I was asked to evaluate the brain SPECT scan images of 20 Veterans pre- and post-hyperbaric oxygen therapy. I was then given an additional 19 Veterans’ brain SPECT scans, for a total of 39 Veterans in this case series. 

4) Additional revisions were made for better clarity.

This manuscript is a resubmission of an earlier submission. The following is a list of the peer review reports and author responses from that submission.

Round 1

Reviewer 1 Report

The manuscript (brainsci-1685179) is a resubmission of a previous one (brainsci-1591955) with minor revision. There is no accompanying letter describing what changes have been made to the manuscript in response to reviewers’ comments. Indeed, major deficits identified in last review were not addressed, including a lack of detail on how the SPECT images were acquired, reconstructed and analyzed, which made the findings difficult to be reproduced or evaluated by another group. Further, no data on MSQ score or other behavior outcomes are presented in the manuscript to support the conclusion that symptoms were improved after HBOT. This makes it impossible to assess the quality of the data and therefore findings. For these critical reasons, the reviewer does not approve acceptance of the manuscript for publication or any further consideration.

Reviewer 2 Report

Reviewer Comments for “A Case Series of 39 United States Veterans with Mild Traumatic Brain Injury Treated with Hyperbaric Oxygen Therapy”

The manuscript “A Case Series of 39 United States Veterans with Mild Traumatic Brain Injury Treated with Hyperbaric Oxygen Therapy” examines the effectiveness of Hyperbaric Oxygen Therapy as a treatment for chronic mild traumatic brain injury within a veteran population. This study demonstrates that following 40 sessions of HBOT, perfusion is increased in the brains of veterans with reported chronic mTBI. The veterans also reported improvement in many symptoms associated with mTBI including sleep, pain, and mood. Interestingly, the veterans included in the study had variable times since injury and location of injury, yet most veterans experience increases in brain perfusion and symptom alleviation after treatment, which supports the idea that HBOT could be an effective therapy for all mTBI populations.

Below are my thoughts and comments I had while reviewing this manuscript separated by headings:

Introduction

  • More discussion is warranted on prior studies investigated HBOT in mild TBI populations. The authors site 5 studies have that have used HBOT in moderate TBI or post-concussive and PTSD populations, however there are several studies that have investigated HBOT within mild TBI populations. A discussion on what these studies found and how the current study is different/adds to literature is needed. Is this the first study to investigate cerebral perfusion pre and post HBOT? Or have prior studies only focuses on symptoms and neurocognitive outcomes and haven’t included an imaging piece? I will refer readers to review the following papers:
    • Ma, J., Hong, G., Ha, E., Hong, H., Kim, J., Joo, Y., Yoon, S., Lyoo, I. K., & Kim, J. (2021). Hippocampal cerebral blood flow increased following low-pressure hyperbaric oxygenation in firefighters with mild traumatic brain injury and emotional distress. Neurological sciences : official journal of the Italian Neurological Society and of the Italian Society of Clinical Neurophysiology42(10), 4131–4138. https://doi.org/10.1007/s10072-021-05094-5
    • Harch, P. G., Andrews, S. R., Rowe, C. J., Lischka, J. R., Townsend, M. H., Yu, Q., & Mercante, D. E. (2020). Hyperbaric oxygen therapy for mild traumatic brain injury persistent postconcussion syndrome: a randomized controlled trial. Medical gas research10(1), 8–20. https://doi.org/10.4103/2045-9912.279978
      • This paper, in particular, has a review table that examines 6 studies that have investigated HBOT in mTBI populations.
    • Hart, B. B., Weaver, L. K., Gupta, A., Wilson, S. H., Vijayarangan, A., Deru, K., & Hebert, D. (2019). Hyperbaric oxygen for mTBI-associated PCS and PTSD: Pooled analysis of results from Department of Defense and other published studies. Undersea & hyperbaric medicine : journal of the Undersea and Hyperbaric Medical Society, Inc46(3), 353–383.
  • The authors discuss the potential mood effects that are often co-occurring with mTBI. However, the authors use the phrase “resemble” anxiety, PTSD, and other psychiatric disorders. I encourage the authors to rephrase this sentence. While mTBI can induce alterations in mood on its own, those who experience mTBI often have comorbid diagnoses of depression, PTSD, and anxiety as well. Using the phrase “resemble” may imply that their symptoms do not qualify for psychiatric diagnoses, when in fact they do.

Methods

  • In the first paragraph of the methods section the authors state that the SPECT scans were “randomly chosen” from veterans who completed the HBOT program. Can you please clarify what this means? Why weren’t all subjects who had SPECT scans completed included in the study?
  • The authors state that the eligibility criteria for the study included a diagnosis of chronic mTBI (between 3-10 years after their mTBI event). Was this a self-reported physician diagnosis? Or was this diagnosis obtained from the veteran’s medical records? Additionally, in the limitations section the authors state that no information was available about the date of injury with mTBI diagnosis. Can you please clarify this? Does this mean all veterans within the study were 3-10 years post mTBI, but information specific to each veteran’s time since injury was not included in the dataset?
  • The authors discuss the major exclusion criteria for HBOT. Do you have information on how many veterans screened out or were unable to undergo HBOT? Many OEF/OIF veterans, in addition to being exposed to mTBI during deployment, we also exposed to numerous neurotoxicant exposures, most prominently burn pits. Several veterans have developed serious lung conditions following these exposures, and thus would not be able to undergo HBOT. Some discussion about this in the limitations section is warranted.
    • Coughlin, S. S., & Szema, A. (2019). Burn Pits Exposure and Chronic Respiratory Illnesses among Iraq and Afghanistan Veterans. Journal of environment and health sciences5(1), 13–14. https://doi.org/10.15436/2378-6841.19.2429
  • The authors state that because of the issues surrounding creating a proper sham or control condition for HBOT, they used a crossover approach. Please explain or define a bit more what that means. By crossover do you mean that you assessed outcomes pre and post HBOT? Or by crossover do you mean that individuals who received HBOT also received a sham/control room-air HBOT condition as well?
  • The behavioral/symptom measurements that were included in the current study, were those only assessed post-treatment? Or were they also assessed pre-treatment? If only post, how did you assess or define “improvement”?
  • Was the dataset received from The 22 Project completely stripped of demographic information? The authors have some information, including age, gender, years from mTBI. If this is individualized data, can the authors please produce a demographic table? Also do the authors have information on co-morbidities in this population? How many of the veterans had PTSD diagnoses or multiple mTBIs?
  • Please explain why you chose not to correct for multiple comparison for this study. If you are assessing ~52 BAs per veteran (39 participants), that would more than 2,000 comparisons. Please consider using a method for controlling for multiple comparisons so that (e.g., FDR, Bonferroni, etc.).

Results

  • The results for the behavioral and symptom assessments are reported in the first paragraph of the results. Were these statistically significant? Can the authors please create a table with the means and SD and p-values for these analyses?
  • The authors state that an improvement in perfusion within a BA was defined as a 10% positive change. Where did this definition come from? Was it from the Paul 2017 paper? Please cite or explain if it is a self-created definition.
  • More explanation is warranted surrounding Table 1. Is this an example of one analysis of one singular BA region? And if so, what region?
  • The authors state that while there was no formal follow-up of participants following the intervention some participant verbally reported that the HBOT benefits have continued. Can the authors please report a specific number of veterans that have reported this? Or a percentage?

Round 2

Reviewer 2 Report

Please see my thoughts and comments on the revised manuscript in the attached word document. 
